# Inter-Fork Strand Annealing causes genomic deletions during the termination of DNA replication

**Carl A Morrow, Michael O Nguyen, Andrew Fower, Io Nam Wong, Fekret Osman, Claire Bryer, Matthew C Whitby\***

Department of Biochemistry, University of Oxford, Oxford, United Kingdom

**Abstract** Problems that arise during DNA replication can drive genomic alterations that are instrumental in the development of cancers and many human genetic disorders. Replication fork barriers are a commonly encountered problem, which can cause fork collapse and act as hotspots for replication termination. Collapsed forks can be rescued by homologous recombination, which restarts replication. However, replication restart is relatively slow and, therefore, replication termination may frequently occur by an active fork converging on a collapsed fork. We find that this type of non-canonical fork convergence in fission yeast is prone to trigger deletions between repetitive DNA sequences via a mechanism we call Inter-Fork Strand Annealing (IFSA) that depends on the recombination proteins Rad52, Exo1 and Mus81, and is countered by the FANCM-related DNA helicase Fml1. Based on our findings, we propose that IFSA is a potential threat to genomic stability in eukaryotes.

## Introduction

Eukaryotic DNA replication initiates at multiple origin sites along each chromosome and terminates when replication forks (RFs) from adjacent origins converge. DNA lesions in the template strands, and physical barriers, such as tightly bound proteins, threaten this seamless progression from initiation to termination by impeding RFs and precipitating their collapse (*Berti and Vindigni, 2016*; *Lambert and Carr, 2013*). RF collapse involves either remodelling or disassembly of the replisome, and can occur with or without DNA breakage (*Berti and Vindigni, 2016*; *Carr et al., 2011*; *Dungrawala et al., 2015*). Regardless of its precise form, collapse renders the fork incompetent for further DNA synthesis. In such cases, DNA replication, initiated from a downstream origin, can ensure that genome duplication is completed (*Mayle et al., 2015*; *Nguyen et al., 2015*; *Yekezare et al., 2013*). However, to guard against the possibility that this second RF may also fail, cells deploy homologous recombination (HR) enzymes to the collapsed fork, which can restart replication (*Lambert et al., 2010*; *Mohebi et al., 2015*; *Nguyen et al., 2015*; *Yeeles et al., 2013*). This so-called recombination-dependent replication (RDR) helps ensure that DNA is fully replicated prior to sister chromatid segregation, thereby avoiding mitotic catastrophes.

The Rad52 protein plays a pivotal role in processing collapsed RFs in both yeast and humans (*Bhowmick et al., 2016*; *Lambert et al., 2010*; *Nguyen et al., 2015*; *Sotiriou et al., 2016*; *Symington et al., 2014*). In yeast, Rad52 mediates the nucleation of the Rad51 recombination protein onto single-stranded (ss) DNA at collapsed forks, which in turn catalyzes strand invasion of homologous DNA to form a displacement (D) loop, at which replication proteins can reassemble to restart DNA synthesis (*Anand et al., 2013*; *Symington et al., 2014*). In addition to aiding Rad51's nucleation onto DNA, Rad52 is also able to anneal complementary ssDNAs coated in Replication protein A (RPA) (*Mortensen et al., 1996*; *Reddy et al., 1997*; *Shinohara et al., 1998*;

**\*For correspondence:** matthew. whitby@bioch.ox.ac.uk

**Competing interests:** The authors declare that no competing interests exist.

*Sugiyama et al., 1998*). This activity promotes several different HR pathways, including a Rad51-independent mode of replication restart (*Anand et al., 2013*; *Symington et al., 2014*). Indeed, Rad51-independent RDR, catalysed by Rad52, appears to be the major pathway in human cells for processing collapsed RFs under conditions of replication stress, and during mitotic prophase (*Bhowmick et al., 2016*; *Sotiriou et al., 2016*). In contrast, yeast mostly use Rad51-dependent RDR, with the Rad52-only pathway seemingly employed as a less efficient alternative (*Anand et al., 2013*; *Lambert et al., 2010*).

RDR can initiate from a collapsed fork where the DNA is either broken or remains intact. Where broken, the exposed DNA is typically resected by a nuclease to generate a ssDNA tail onto which first RPA and then Rad52 loads, leading to a version of RDR called break-induced replication (BIR) (*Anand et al., 2013*). However, where the collapsed RF remains unbroken, restart is thought to involve its regression, where the parental DNA strands re-anneal and cause the nascent strands to unwind and base-pair with each other, forming a free double-stranded end (*Atkinson and McGlynn, 2009*; *Sun et al., 2008*). Nucleolytic resection of this end generates the requisite ssDNA tail onto which the recombination proteins can load and drive the restart process (*Berti and Vindigni, 2016*).

In yeast, the loading of Rad52 at DSBs and collapsed RFs takes approximately 10 min, but DNA synthesis takes even longer to start (*Hicks et al., 2011*; *Nguyen et al., 2015*). This means that DNA replication will frequently complete by an active fork, initiated from a downstream replication origin, converging on a collapsed fork (*Nguyen et al., 2015*). Completing DNA replication in this manner is thought to be desirable, because it prevents the risk of chromosomal rearrangements and copy-number variations (CNVs), which can be caused by the restart process inadvertently recombining repetitive DNA elements both at, and downstream of, the collapsed fork (*Mayle et al., 2015*; *Nguyen et al., 2015*). However, it is unknown whether this type of replication termination, where an active fork converges on a collapsed fork, can itself cause genome instability.

In this study we show that convergence of an active fork with a collapsed fork can result in surprisingly high rates of inter-DNA repeat recombination leading to genomic deletions. Three key recombination proteins that drive deletion formation, including Rad52, are identified, as well as one that actively suppresses it. We propose a mechanism to explain how the deletions are formed, and speculate that it may be responsible for some of the CNVs that give rise to disease in humans.

## Results

### Experimental system

To investigate how RF collapse influences inter-repeat recombination, we used the site-specific RF barrier *RTS1* to block forks at the *ade6* locus on chromosome 3 in fission yeast. *RTS1* is a unidirectional barrier, formed by a myb domain-containing DNA binding protein called Rtf1 (*Vengrova et al., 2002*). Due to the position of flanking replication origins, the direction of replication at the *ade6* locus is strongly polarized in the telomere to centromere direction (*Figure 1A*). Consequently only one orientation of *RTS1* blocks RFs at this site, and this is referred to as the active orientation (AO). The opposite orientation, that does not block RFs, is called the inactive orientation (IO). By live cell imaging, we have shown that recombination protein foci are recruited to *RTS1*-AO within approximately 10 min of RF blockage (*Nguyen et al., 2015*). DNA repeat recombination is measured in cells where *RTS1* is flanked by a pair of directly oriented *ade6*⁻ heteroalleles approximately 6 kb apart, which recombine to give either an *ade6*⁺ gene conversion or deletion (*Figure 1A*). With *RTS1*-AO we observe a > 50 fold increase in inter-repeat recombination compared to cells without *RTS1* or with *RTS1*-IO, and the ratio of deletions and gene conversions is approximately 40:60 (*Figure 1B*) (*Nguyen et al., 2015*). The major recombination proteins Rad51 and Rad52 are both required for gene conversion, whereas Rad52 can form approximately 40–50% of deletions without Rad51 (*Ahn et al., 2005*; *Lorenz et al., 2009*; *Nguyen et al., 2015*; *Sun et al., 2008*).

### Increasing the distance between DNA repeats can dramatically increase the frequency of deletions

To investigate how the proximity of DNA repeats to the site of RF collapse influences inter-repeat recombination, we systematically added increasing lengths of spacer DNAs either side of *RTS1*-AO

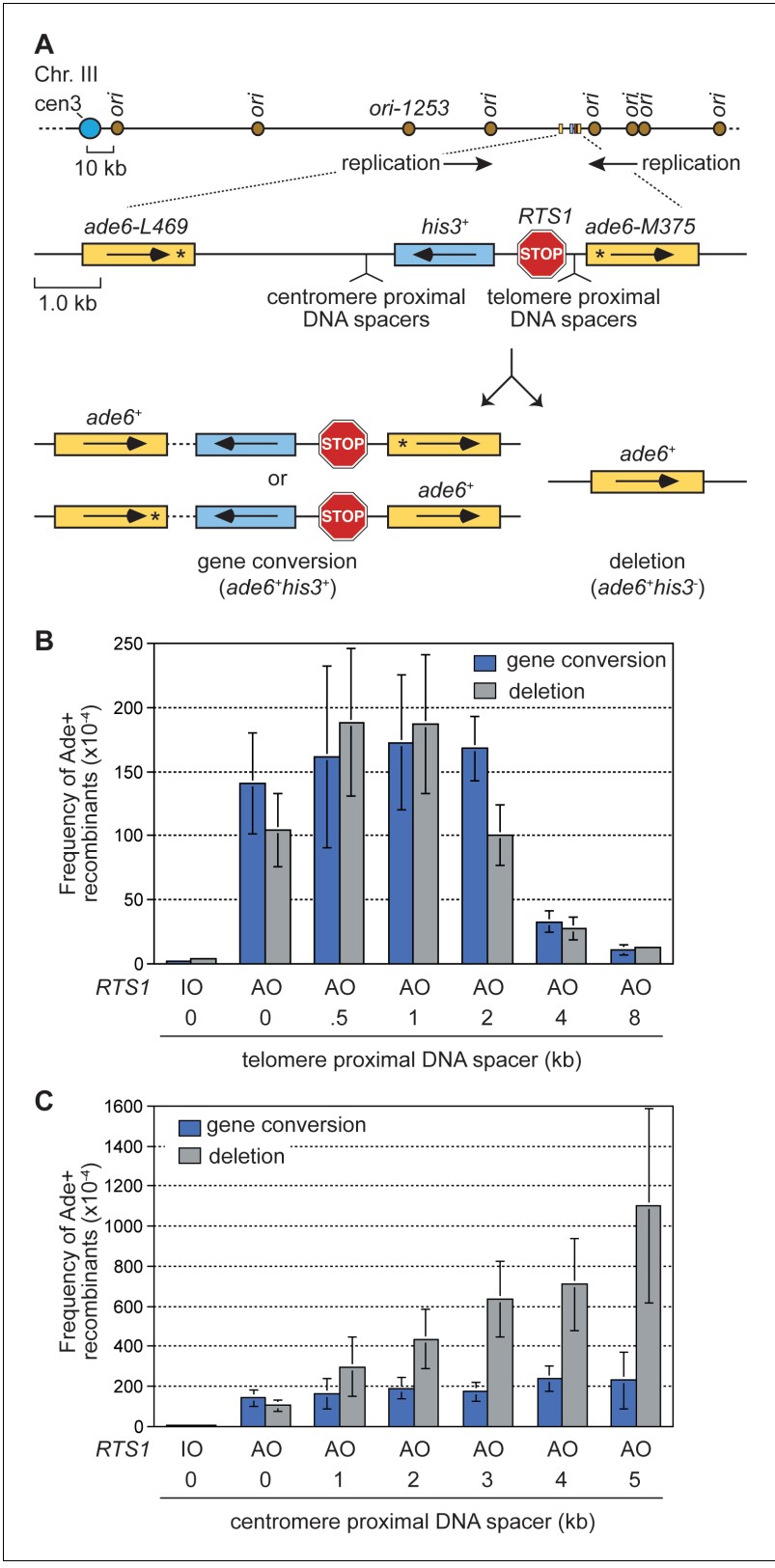

**Figure 1.** The effect of inter-DNA repeat distance on RF barrier-induced recombination. (**A**) Schematic showing the location of *RTS1*, *ade6-* direct repeats, sites of DNA spacer insertion, nearby replication origins, and products of inter-repeat recombination. Asterisks indicate the position of point mutations in the *ade6* alleles. (**B** to **C**) Mean
*Figure 1 continued on next page*

*Figure 1 continued*

frequency of Ade+ gene conversions and deletions in wild-type cells with *RTS1*-IO or -AO and different length telomere- and centromere-proximal DNA spacers. Error bars represent 1 SD.

The following source data is available for figure 1:

**Source data 1.** Effect of inter-repeat distance on the frequency of *RTS1*-AO-induced direct repeat recombination.

to expand the distance between the *ade6⁻* heteroalleles (**Figure 1A**). Modest increases in inter-repeat recombination were observed with the addition of 0.5 and 1 kb spacers on the telomere-proximal side of *RTS1*-AO, whereas 4 and 8 kb spacers resulted in a 4- and 10-fold reduction, respectively (**Figure 1B**). This reduction in inter-repeat recombination is consistent with ChIP data showing that Rad52 enrichment on DNA extends only up to ~2 kb behind RFs blocked at *RTS1* (**Tsang et al., 2014**). In stark contrast, the addition of DNA spacers on the centromere-proximal side of *RTS1*-AO resulted in a dramatic increase in inter-repeat recombination, rising from an ~1.9 fold increase, with a 1 kb spacer, to ~5.4 fold with a 5 kb spacer (**Figure 1C**). Importantly, no increase in spontaneous inter-repeat recombination was observed with the 5 kb spacer (**Figure 1—source data 1**). Crucially, most of the extra recombinants (~90%) were Rad51-independent deletions, which depended on Rad52 (**Figure 2**). Indeed the frequency of these deletions increased by ~10 fold with the addition of the 5 kb spacer (**Figure 1C**). Henceforth, we will refer to these extra deletions, which are observed when DNA spacers are inserted on the centromere-proximal side of *RTS1*-AO, as spacer-dependent deletions (SDDs).

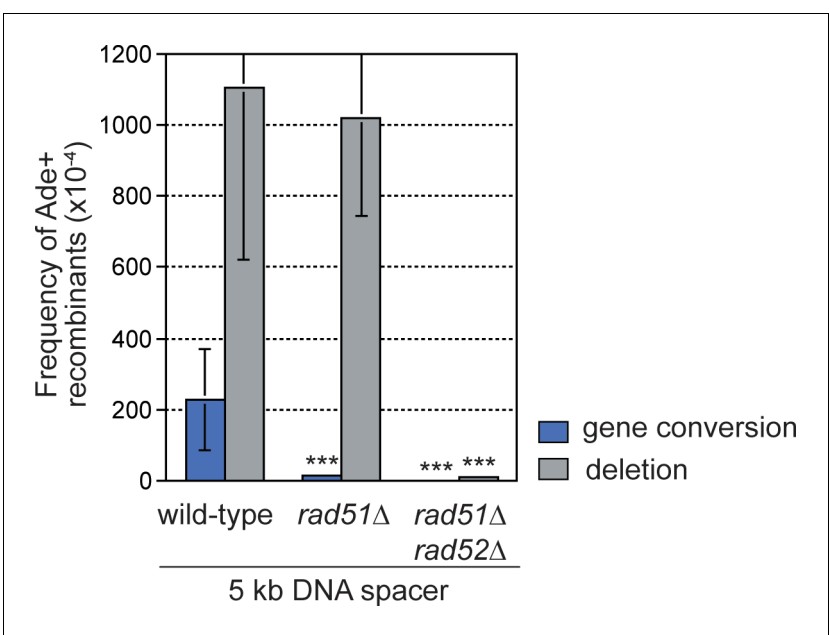

**Figure 2.** DNA spacer-dependent deletions depend on Rad52 but not Rad51. Mean frequency of Ade+ gene conversions and deletions in wild-type, *rad51Δ* and *rad51Δ rad52Δ* strains containing *RTS1*-AO with a 5 kb centromere-proximal DNA spacer. Significant fold changes compared to the wild-type strain are indicated (***p<0.001). Error bars represent 1 SD.

The following source data is available for figure 2:

**Source data 1.** Frequency of *RTS1*-AO-induced direct repeat recombination in wild-type, *rad51Δ* and *rad51Δ rad52Δ* strains with an extra 5 kb DNA spacer between the repeats.

## RF convergence is needed for SDDs

Data, from live cell imaging experiments, show that both the occurrence and duration of Rad52 foci at *RTS1*-AO are restricted by RF convergence (*Nguyen et al., 2015*). Indeed, ~40% of cells fail to show a Rad52 focus at *RTS1*-AO during their cell cycle due to the centromere-proximal RF converging on the blocked fork prior to Rad52 recruitment (*Nguyen et al., 2015*). Deletion of the closest major replication origin (*ori1253*), on the centromere-proximal side of *RTS1*-AO (*Figure 1A*), results in the incoming RF having to travel from a more distant origin in most cells, allowing more time for replication restart (*Nguyen et al., 2015*). Indeed, deletion of *ori1253* results in: (1) almost all cells exhibiting a Rad52 focus at *RTS1*-AO (*Nguyen et al., 2015*); (2) a > 2 fold increase in inter-repeat recombination in strains with no additional DNA spacer (*Figure 3A*); and (3) a greater number of forks restarting and moving away from *RTS1*-AO, as evidenced by increased template switching downstream of the barrier (*Nguyen et al., 2015*). Crucially, the increased frequency of SDDs, observed in strains containing the 5 kb DNA spacer, is reduced by ~2.3 fold (p=<0.001) when *ori1253* is deleted (*Figure 3A*), suggesting that RF convergence is necessary for the increase.

To verify that delaying the incoming RF results in a reduction in SDDs, we placed tandem repeats of three *Ter2/3* barriers downstream of *RTS1*-AO to block the incoming fork (*Figure 3—figure supplement 1*). *Ter2/3* is a unidirectional barrier but, unlike *RTS1*, it does not trigger recombination (*Mizuno et al., 2013*). First, we determined what affect *Ter2/3* has on Rad52 focus co-localization with *RTS1*-AO (*Figure 3—figure supplement 1*). Similar to deleting *ori1253*, *Ter2/3* increased both the occurrence and duration of Rad52 foci at the barrier (*Figure 3—figure supplement 1*, and data not shown). Next, we determined whether *Ter2/3* affected the frequency of inter-repeat

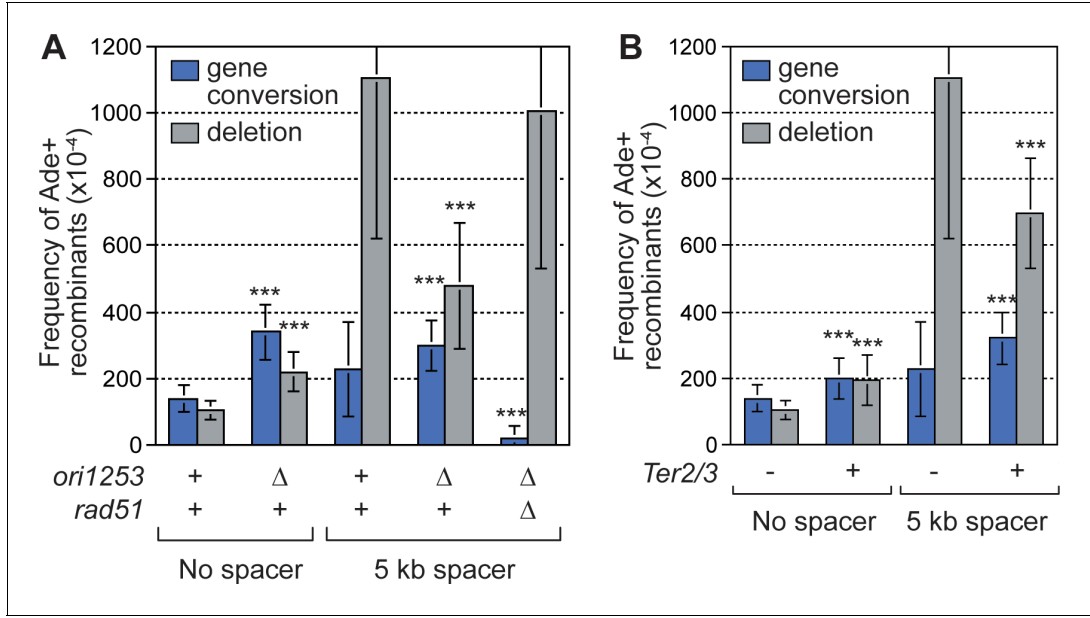

**Figure 3.** Deletion of the downstream replication origin *ori1253*, or introduction of a *Ter2/3* barrier downstream of *RTS1*-AO, reduces the frequency of SDDs. (A) Mean frequency of Ade+ gene conversions and deletions in wild-type and *ori1253Δ* strains containing *RTS1*-AO with and without a 5 kb centromere-proximal DNA spacer. The effect of deleting *rad51* on the frequency of recombinants in a *ori1253Δ* strain with a 5 kb centromere-proximal DNA spacer is also shown. (B) Effect of a *Ter2/3* barrier downstream of *RTS1*-AO on the frequency of Ade+ gene conversions and deletions in wild-type strains with and without a 5 kb centromere-proximal DNA spacer. Significant fold changes compared to the equivalent wild-type strain are indicated (***p<0.001). Error bars represent 1 SD.

The following source data and figure supplement are available for figure 3:

**Source data 1.** Effect of delaying fork convergence on the frequency of *RTS1*-AO-induced direct repeat recombination with and without an extra 5 kb DNA spacer between the repeats.
**Figure supplement 1.** The effect of delaying the incoming RF, with a *Ter2/3* barrier, on recruitment of Rad52 to *RTS1*.

recombination induced by *RTS1*-AO (*Figure 3B*). With no additional DNA spacer between the repeats, the inclusion of *Ter2/3* downstream of *RTS1*-AO resulted in ~1.4 fold increase in gene conversions (p=<0.001) and ~1.9 fold increase in deletions (p=<0.001). However, the elevated frequency of SDDs, observed with the additional 5 kb DNA spacer, was reduced by ~1.6 fold (p=<0.001) (*Figure 3B*).

The aforementioned data show that delaying the incoming RF, either by deleting *ori1253* or adding *Ter2/3*, suppresses SDDs, despite overall recombination activity at *RTS1*-AO increasing. This strongly indicates that SDDs are promoted by RF convergence rather than replication restart. If true, we reasoned that reducing the efficiency of replication restart should counteract the effect of *ori1253* deletion in suppressing SDDs. As Rad51 is needed for efficient replication restart (*Lambert et al., 2010*), we determined what effect its mutation has on SDDs in an *ori1253Δ* background (*Figure 3A*). In accordance with our prediction, a *rad51Δ* mutant suppressed the reduction in SDDs observed when *ori1253* is deleted. These data provide further support for the notion that RF convergence at *RTS1*-AO is needed to promote SDDs.

## Factors that promote or prevent SDDs

To identify recombination proteins that either promote or prevent deletion formation during RF convergence, we investigated candidates including Exonuclease 1 (Exo1), Mus81-Eme1 and Fml1. Exo1 promotes recombination by catalyzing 5′ to 3′ DNA strand resection (*Keijzers et al., 2016*), and its loss results in a ~2.5 fold reduction in gene conversions, and a ~2 fold reduction in deletions, in strains with *RTS1*-AO and no additional DNA spacer (*Figure 4A*) (*Osman et al., 2016*). Its loss also causes a similar reduction in gene conversions (~2.2 fold) when the centromere-proximal 5 kb spacer is added (*Figure 4A*). However, SDDs are reduced by ~4.2 fold, indicating that DNA resection plays an especially important role in their formation (*Figure 4A*).

The heterodimeric structure-specific DNA endonuclease Mus81-Eme1 can cleave both stalled RFs and recombination intermediates, including D-loops and Holliday junctions (*Osman and Whitby,*

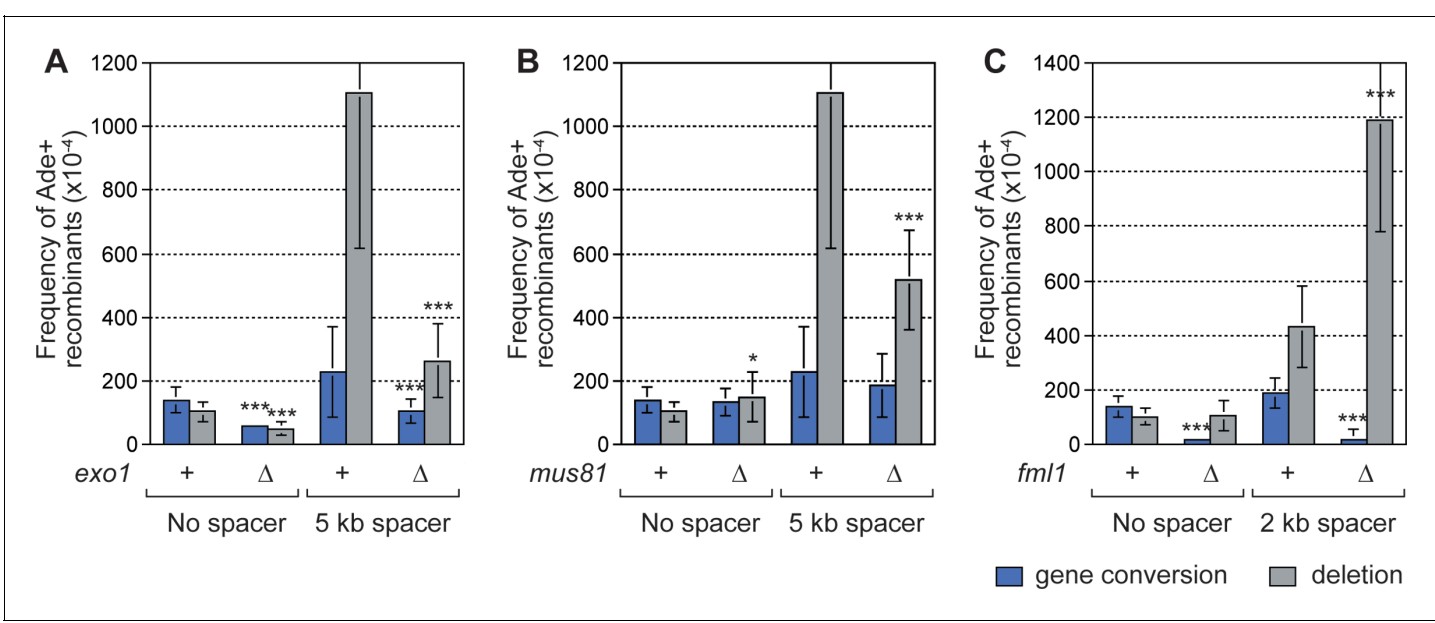

**Figure 4.** SDDs depend on Exo1 and Mus81, but are suppressed by Fml1. (**A–C**) Mean frequency of Ade+ gene conversions and deletions in wild-type and mutant strains containing *RTS1*-AO with and without a centromere-proximal DNA spacer as indicated. Significant fold changes compared to the equivalent wild-type strain are indicated (*p<0.05, **p<0.01, ***p<0.001). Error bars represent 1 SD.

The following source data is available for figure 4:

**Source data 1.** Frequency of *RTS1*-AO-induced direct repeat recombination in wild-type, *exo1Δ*, *mus81Δ* and *fml1Δ* strains with and without an extra DNA spacer between the repeats.

*2007*; *Rass, 2013*; *West et al., 2015*). However, a *mus81Δ* mutant has little or no effect on the frequency of inter-repeat recombination, induced by RF blockage at *RTS1*-AO, when there is no additional DNA spacer between the repeats (*Figure 4B*). In contrast, loss of *mus81* causes a > 2 fold reduction in SDDs in strains with either a 2 or 5 kb centromere-proximal spacer DNA (*Figure 4B*), indicating that Mus81-Eme1 specifically promotes SDDs.

Fml1 is a member of a conserved family of DNA helicases, which includes human FANCM, that have been implicated in processing stalled RFs and D-loops (*Whitby, 2010*; *Xue et al., 2015*). In fission yeast, Fml1 promotes Rad51-dependent gene conversion in response to RF blockage at *RTS1*-AO, but has little effect on the frequency of DNA deletions when there is no additional DNA spacer between the repeats (*Figure 4C*) (*Sun et al., 2008*). In contrast, loss of *fml1* causes a ~2.7 fold increase in SDDs over the wild-type level, in strains with a 2 kb centromere-proximal DNA spacer (*Figure 4C*), indicating that Fml1 limits SDDs.

## A new model: Inter-Fork Strand Annealing

Rad52 can repair DSBs by catalyzing single-strand annealing (SSA) between directly oriented repeats and, in so-doing, causes DNA deletions (*Bhargava et al., 2016*). However, in wild-type cells, RFs blocked at *RTS1*-AO are not broken (*Mizuno et al., 2009*; *Nguyen et al., 2015*). Moreover, expansion of the inter-repeat distance does not increase SSA. On the contrary, it results in a greater proportion of DSBs being repaired by gene conversion (*Fishman-Lobell et al., 1992*; *Schildkraut et al., 2005*). We propose a new model, called Inter-Fork Strand Annealing (IFSA), to explain how Rad52 could promote deletions in the absence of RF breakage (*Figure 5*). Following RF blockage and collapse at *RTS1*-AO, the fork regresses exposing a DNA end that is resected by Exo1 (*Figure 5*, step 2). Resection generates a ssDNA tail, incorporating the *ade6⁻* repeat on the telomere proximal side of the barrier, which Rad52 binds to (*Figure 5*, step 3). During RF convergence, Rad52 inadvertently anneals its *ade6* repeat to the complementary ssDNA of the other *ade6* repeat that is exposed in the lagging strand gap of the incoming fork (*Figure 5*, step 4). Strand annealing would be aided by the DNA repeats being closely juxtaposed in three-dimensional space, which may be facilitated by the addition of the DNA spacer. Inter-fork strand annealing establishes a connection between the two RFs bounded by three/four-way DNA junctions at either end of the annealed DNA. Appropriate cleavage of this 'IFSA junction' by Mus81-Eme1 would generate one intact sister chromatid and one with the DNA repeat region deleted (*Figure 5*, steps 4 and 5). In vitro, Fml1 can catalyse RF regression (*Nandi and Whitby, 2012*; *Sun et al., 2008*), and also re-set regressed forks (our unpublished data). Therefore, it may limit deletion formation by modulating the extent of RF regression (*Figure 5*,

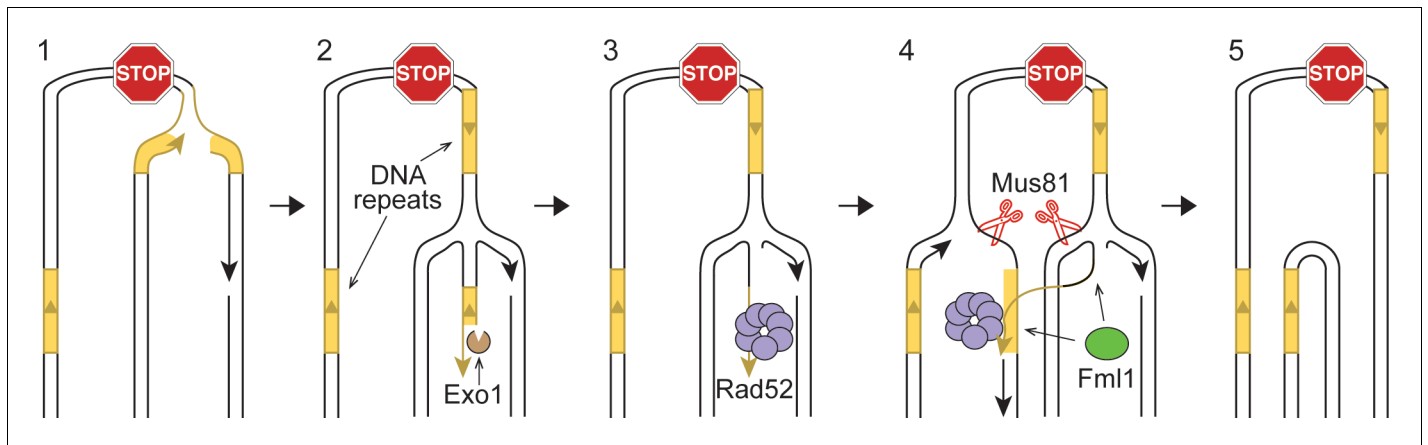

**Figure 5.** IFSA model. See main text for details.
The following figure supplement is available for figure 5:

**Figure supplement 1.** Alternative models for IFSA.

step 4). Alternatively it might unwind the annealed strands prior to IFSA junction cleavage (*Figure 5*, step 4).

Whilst we favour the idea that RF regression is key to converting the stalled fork into a substrate for recombination, there are alternative ways by which IFSA may initiate (*Figure 5—figure supplement 1*). For example, instead of RF regression, the leading nascent DNA strand may simply be unwound to generate a 3'-ended ssDNA tail that Rad52 binds (*Figure 5—figure supplement 1A*). Alternatively, Exo1 may expand the lagging strand gap at the stalled fork, exposing ssDNA, which again could be bound by Rad52. IFSA may then occur through annealing of the lagging strand gaps (*Figure 5—figure supplement 1B*).

## Mus81-Eme1 resolves a model IFSA junction

Based on previous in vitro experiments, we knew that Mus81-Eme1 had the potential to cleave the different versions of the IFSA junction in the manner predicted by our models (*Figure 5* and *Figure 5—figure supplement 1*) (*Whitby et al., 2003*). To confirm this for the junction predicted by our favoured model (*Figure 5*), we established an in vitro reaction using purified recombinant Mus81-Eme1 and a synthetic IFSA junction. Native gel analysis of the reaction products showed that Mus81-Eme1 could indeed resolve an IFSA junction into two nicked/gapped linear duplex DNA products, whereas a catalytically inactive mutant could not (*Figure 6A*). Mapping the cleavage sites on a denaturing gel showed that Mus81-Eme1 made incisions in strand 1 close to both four-way DNA junction points, which are consistent with those predicted by our model (*Figure 6B and C*). An additional minor cleavage site in strand 2 was also detected (*Figure 6B and C*).

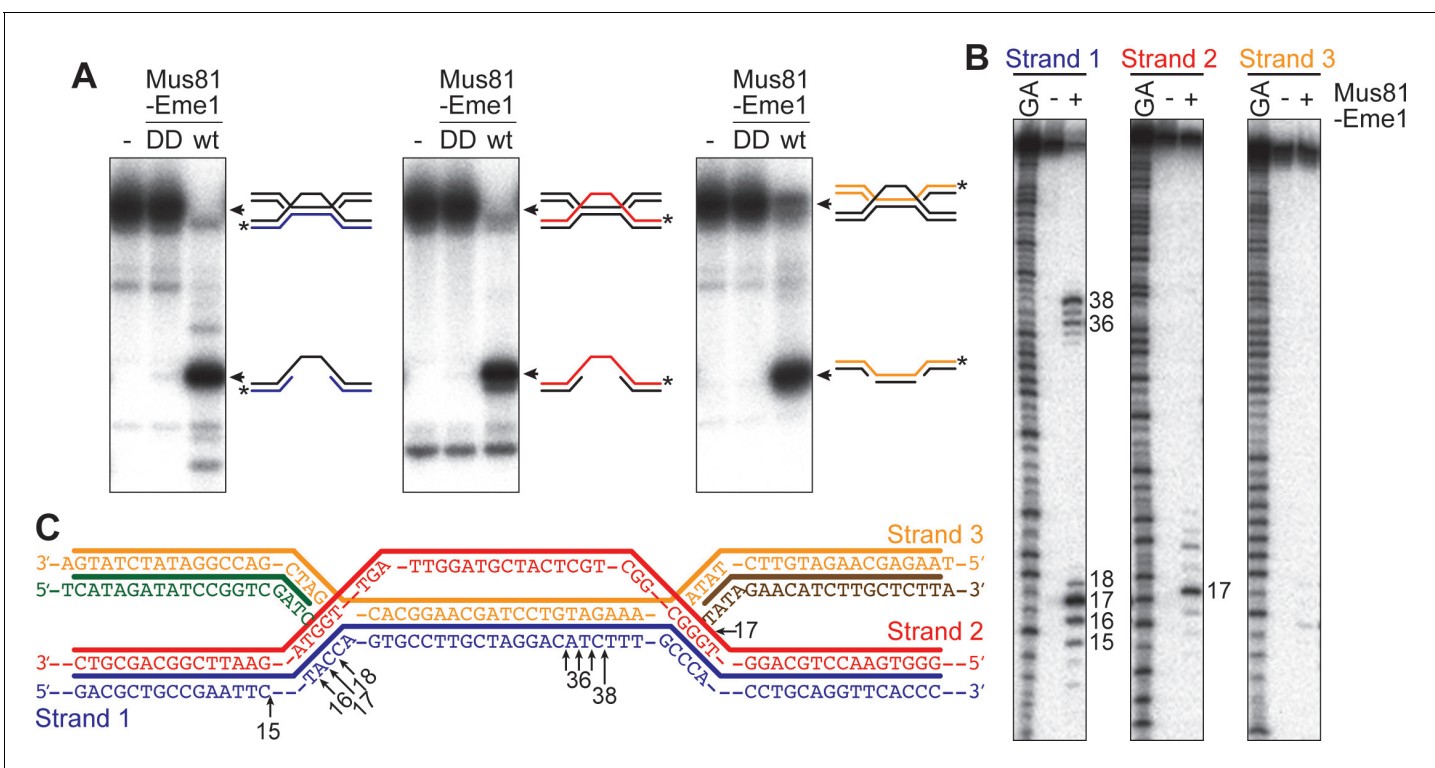

**Figure 6.** Cleavage of a model IFSA junction by Mus81-Eme1. (**A**) Native gel analysis of Mus81-Eme1 cleavage reactions with wild-type (wt) and catalytically inactive (DD) enzyme. The schematics show the IFSA junction and cleavage products. The asterisks indicate the position of the 5'-end [32]P label. (**B**) Denaturing gel analysis of Mus81-Eme1 cleavage reactions to map the sites of strand cleavage by reference to a G and A Maxam-Gilbert sequencing ladder of the relevant labelled strand. (**C**) Schematic showing the sites of strand cleavage (arrows).

# Discussion

We have shown that non-canonical RF convergence, involving a regular fork converging on a collapsed fork, can promote recombination between DNA repeats that flank the site of replication termination by a process we call IFSA, which involves Rad52-mediated strand annealing between the two RFs. Previously, we, and others, have reported that RDR, promoted by fork collapse at the *RTS1* barrier, generates chromosomal rearrangements and CNVs, both during the act of restarting replication, and from the increase in template switching associated with the restarted fork (*Ahn et al., 2005*; *Lambert et al., 2010*, *2005*; *Mizuno et al., 2009*, *2013*; *Nguyen et al., 2015*). However, IFSA is distinct from previously described forms of *RTS1*-induced recombination due to its dependence on RF convergence. Indeed, this requirement contrasts markedly with RDR-associated template switching, which is constrained by RF convergence (*Mayle et al., 2015*; *Nguyen et al., 2015*). It is also distinct from other types of replication slippage event, which typically envisage the 'slippage' occurring between DNA repeats at the same RF rather than between converging RFs (*Lovett, 2004*).

Through the application of microarray and next-generation sequencing technologies, there has been a growing appreciation of the prominent role that CNVs play in human genetic diversity and disease (*Stankiewicz and Lupski, 2010*). For example, chromosomal deletions arising in germ cells have been linked to a variety of genetic disorders, including complex syndromes such as autism and schizophrenia, whilst their occurrence in somatic cells has been associated with cancer (*Beroukhim et al., 2010*; *Chen et al., 2010*; *Kim et al., 2013*; *Watson et al., 2014*). The formation of CNVs and other types of structural variants (SVs) in humans has been attributed to three main types of mechanism: non-allelic homologous recombination (NAHR); non-homologous end joining (NHEJ); and replication based mechanisms, such as fork stalling and template switching (FoSTeS), and microhomology-mediated break-induced replication (MMBIR) (*Carvalho and Lupski, 2016*). The class of mechanism responsible for any given CNV or SV is typically inferred by characterization of the breakpoint junctions, with the presence of long stretches of homologous sequence signifying NAHR, little or no homology indicating NHEJ, and microhomology, together with a short inserted sequence, pointing towards a replication based mechanism (*Carvalho and Lupski, 2016*; *Weckselblatt and Rudd, 2015*).

In our experimental system, IFSA occurs between directly orientated DNA repeats that share ~1.7 kb of almost perfect homology and, as such, would be classed as a type of NAHR. NAHR is generally thought to occur when ectopic homologous sequences (typically low-copy repeats that are >10 kb in size and share >95–97% sequence identity) are inadvertently aligned during Rad51 family recombinase-mediated DSB repair, in which intermediates are resolved to give a crossover outcome (*Carvalho and Lupski, 2016*; *Chen et al., 2010*). Indeed the majority of recurrent SVs, associated with known genomic disorders, are most likely formed during the repair of meiotic DSBs (*Carvalho and Lupski, 2016*; *Chen et al., 2010*). Could IFSA be responsible for a subset of SVs that have been attributed to homology-directed DSB repair? The following observations encourage us to think that it might. Firstly, the principle players that direct IFSA in *S. pombe*, Rad52 and Mus81, are not only conserved in humans but also act in the same pathway to process collapsed RFs and, therefore, are likely to be present at sites of non-canonical RF convergence (*Bhowmick et al., 2016*; *Sotiriou et al., 2016*). Secondly, many of the documented SVs are simple deletions formed by recombination between either low-copy repeats, or far more abundant transposon sequences, such as long interspersed nuclear elements (LINEs) and Alu elements (*Carvalho and Lupski, 2016*; *Flynn et al., 2014*; *Kim et al., 2016*; *Startek et al., 2015*). Collectively, these repetitive sequences make up ~50% of the human genome and, therefore, there are likely to be occasions when non-canonical RF convergence occurs in the vicinity of sites that can be annealed. Thirdly, inter-repeat recombination is implicated in the formation of a growing number of disease-associated non-recurrent SVs, including those where there is no enrichment for the PRDM9 meiotic DSB hotspot motif (*Carvalho and Lupski, 2016*). This highlights the possibility that some SVs, associated with DNA repeats, are formed by mechanisms other than Rad51 family recombinase-mediated meiotic DSB repair. Indeed, it has been proposed that many of the non-recurrent SVs that are associated with Alu elements are formed by FoSTeS or MMBIR, since these elements can share as little as 75% identity and the SV breakpoint junctions often exhibit only microhomology (*Carvalho and Lupski, 2016*). Currently, we do not know what tolerance IFSA has for divergent repeat sequences and, therefore,

it is unclear whether it could contribute to deletions formed between Alu elements, such as those responsible for the genetic disorders neurofibromatosis type one and autosomal-dominant spastic paraplegia 4 (*Boone et al., 2014*; *Hsiao et al., 2015*). However, in a DSB-induced chromosomal translocation assay in mouse cells, SSA accounted for 85% of translocations when identical Alu elements were adjacent to the DSBs, but only 4% when the Alu elements diverged by 20% (*Elliott et al., 2005*). Whether IFSA is similarly affected by divergence of DNA repeat sequences remains to be determined.

In mammals, the average distance between active replication origins during S phase is ~150 kb (*Takebayashi et al., 2017*) and, therefore, this would limit the size of chromosomal deletions that IFSA could generate if it only occurs during RF convergence. However, we suspect that IFSA might act between any collapsed and active RF that share a repetitive sequence, as long as they are in close proximity with each other (e.g. RFs in the same replication factory). If true, this would provide additional scope for generating larger deletions and even other types of SV. For example, strand annealing between RFs travelling in the same direction through inverted repeats could result in the formation of an additional acentric or dicentric chromosome, with the former predisposed to micronuclei formation (*Fenech et al., 2011*), and the latter capable of causing breakage-fusion-bridge (BFB) cycles (*Matsui et al., 2013*). Both of these events are thought to drive tumorigenesis, either by promoting chromothripsis (in the case of micronuclei formation), or by oncogene amplification (in the case of BFB cycles) (*Leibowitz et al., 2015*; *Matsui et al., 2013*).

In some respects, IFSA is reminiscent of models, such as FoSTeS and MMBIR, that have been proposed to explain stress-induced gene amplification in bacteria, and various complex non-recurrent genome rearrangements in humans, which similarly invoke the concept of strand annealing between RFs, where one of the two RFs is stalled or broken (*Hastings et al., 2009*; *Lee et al., 2007*; *Slack et al., 2006*). In the case of MMBIR, it has also been suggested that Rad52 promotes the strand annealing step (*Hastings et al., 2009*). However, the genomic rearrangements that these models describe are characterised by microhomology (typically 2–33 bp in humans) at the breakpoint junctions rather than the ~1.7 kb regions of homology used in our experimental system (*Carvalho and Lupski, 2016*). Moreover, FoSTeS and MMBIR depend on template switching events, which can prime DNA synthesis and lead to the copying of nearby genomic sequences that are found inserted at the breakpoint junctions (*Carvalho and Lupski, 2016*). In contrast, our model for IFSA does not invoke a requirement for DNA synthesis following strand annealing. However, it has been suggested that FoSTeS/MMBIR may also account for many simple non-recurrent SVs, including interstitial deletions (*Carvalho and Lupski, 2016*; *Hastings et al., 2009*). Such deletions could also be formed during DSB repair by so-called microhomology-mediated end joining (MMEJ – also known as alternative end joining) (*Hastings et al., 2009*). Indeed, in one study of somatic SVs in multiple cancer types, 41% of deletions were attributed to MMEJ based on the presence of between 2–40 bp of microhomology at their breakpoint junctions (*Yang et al., 2013*). Whilst we have not demonstrated that IFSA can function when there is only microhomology shared between the converging RFs, we suspect that it might be responsible for some of the deletions documented in cancer genomes, which have hitherto been attributed to MMEJ or FoSTeS/MMBIR. Our suspicion is based on: (1) knowledge that the genomic regions marking the boundaries of somatic CNVs found in many cancers tend to be replicated at the same time and share a long-range interaction (*De and Michor, 2011*); (2) reports that Rad52 promotes DSB repair by MMEJ in fission yeast, and during antibody class-switch recombination in mouse B cells (*Decottignies, 2007*; *Zan et al., 2017*), albeit the same may not be true in budding yeast (*Meyer et al., 2015*). If Rad52 is capable of annealing DNA strands that share only microhomology at a DSB, then it might also be able to do the same to strands exposed during non-canonical RF convergence.

Our discovery that non-canonical RF convergence can result in chromosomal deletions by IFSA highlights a potential novel risk to genomic stability. However, whether it is likely to be a significant contributor to pathogenic CNVs in humans will depend on how efficiently it can operate when sequence homology between the converging RFs is limited. Finally, it is interesting to speculate on whether IFSA might serve a beneficial function to the cell similar to SSA, which promotes cell survival by repairing DSBs at the expense of generating DNA deletions (*Bhargava et al., 2016*). In this regard, we imagine that IFSA may aid the termination of DNA replication at specific problem sites, enabling sister chromatids to segregate successfully during mitosis. This possibility warrants investigation in future studies.

## Materials and methods

### Yeast strains, plasmids and PCR primers

*Schizosaccharomyces pombe* strains and PCR primers used for this study are listed in *Supplementary files 1* and *2*, respectively. Plasmids pCB29, pCB31, pCB33, pAF6 and pAF7 were used for the construction of strains with *RTS1-AO* flanked by a direct repeat of *ade6-L469* and *ade6-M375* with varying sized DNA spacers between *RTS1-AO* and *ade6-M375*. They are derivatives of pCB24, which in turn is a derivative of pFOX2 (*Osman et al., 2000*) containing *RTS1-AO* inserted at its SalI site. *RTS1-AO* was amplified from pMW700 (*Ahn et al., 2005*) using PCR primers oMW1556 and oMW1557. pCB29 contains a ~0.5 kb DNA spacer, amplified from pACYC184 using PCR primers oMW1558 and oMW1560, and inserted between the SpeI and XmaI sites in pCB24. pCB31 and pCB33 were constructed in the same way as pCB29 but with ~1 kb and ~2 kb DNA spacers, respectively. These DNA spacers were amplified from pACYC184 using primer oMW1558 with either oMW1561 (1 kb) or oMW1562 (2 kb). pAF6 was also constructed in the same way as pCB29, but its DNA spacer is ~8 kb and was amplified from Bacteriophage Lambda DNA (New England BioLabs UK Ltd) using primers oMW1643 and oMW1644. pAF7 is a derivative of pAF6 with the ~3.9 kb SexAI fragment within the ~8 kb DNA spacer removed. Plasmids pAF8, pAF10, pAF11, pAF12 and pAF13 were used for the construction of strains with *RTS1-AO* flanked by a direct repeat of *ade6-L469* and *ade6-M375* with varying sized DNA spacers between *RTS1-AO* and *ade6-L469*. pAF8 was made by first amplifying a ~1 kb fragment from Lambda DNA using oMW1700 and oMW1701, and then inserting this at the PvuII site closest to *his3* in pCB24. pAF10, pAF11, pAF12 and pAF13 are all derivatives of pAF8 with the ~1 kb spacer DNA replaced with a larger sized spacer amplified from Lambda DNA. These larger DNA spacers were amplified using primer oMW1700 together with one of the following primers: oMW1712 (2 kb DNA spacer); oMW1713 (3 kb DNA spacer); oMW1714 (4 kb DNA spacer); and oMW1715 (5 kb DNA spacer). pAF16 was used for the construction of strains with *RTS1-IO* flanked by a direct repeat of *ade6-L469* and *ade6-M375* with a ~5 kb DNA spacer between *RTS1-IO* and *ade6-L469*. It was made by excising *RTS1* from pAF13 by SalI digestion, and then reinserting it in the opposite orientation. Strains containing the *ade6*⁻ direct repeat flanking *RTS1*, with the various sized DNA spacers, were constructed by transforming the appropriate BlpI-linearized plasmid (i.e. pAF16, pCB24 or pCB24 derivative) into a *ade6-M375* strain (*Osman et al., 2000*). pCB41 is a derivative of pAG25 (*Goldstein and McCusker, 1999*) used for the targeted insertion of 3 tandem repeats of the ribosomal DNA fork barrier *Ter2/3* (*Mizuno et al., 2013*) between *srk1* and *SPCC1322–09* ~10 kb upstream of *ade6-L469*. All plasmids were verified by DNA sequencing. All strains were verified by confirming genetic markers through a combination of phenotypic tests, assessing auxotrophy, prototrophy, antibiotic resistance and genotoxin sensitivity, and physical analysis of the DNA by PCR, DNA sequencing and Southern blotting as necessary.

### Recombination assays

Direct repeat recombination was assayed by measuring the frequency of Ade⁺ recombinants as described (*Osman and Whitby, 2009*). Between 3 to 10 colonies were assayed in each experiment, with experiments repeated at least three times to achieve a minimum sample size as calculated using the Power calculation $n = f(\alpha,\beta)(2 \ s^2/\delta^2)$ where $\alpha = 0.05$; $\beta = 0.1$; $s = 40$; and $\delta = 50$. Strains being directly compared were analysed at the same time in parallel experiments. Statistical analysis of the recombination data was performed in SPSS Statistics Version 22 (IBM, Armonk, NY). Each data set was tested for normal distribution using a Shapiro-Wilk test, rejecting the null hypothesis (H₀; 'data fits a normal distribution') at an $\alpha$-level of $p<0.05$. Several data sets did not conform to a normal distribution and, therefore, all comparisons were done using a two-tailed, two independent sample Wilcoxon rank-sum test (also known as the Mann-Whitney U test). This test is non-parametric and does not depend on data sets being normally distributed. Relevant *p* values are given in *Figure 1—source data 1*, *Figure 2—source data 1*, *Figure 3—source data 1* and *Figure 4—source data 1*.

### Microscopy and image analysis

Live cell imaging and image analysis of Rad52-YFP and LacI-tdKatushka2 foci has been described previously (*Nguyen et al., 2015*). The data for each strain in *Figure 3— figure supplement 1* is derived from at least two independent cell cultures.

## Synthetic IFSA junction

The IFSA junction was made by annealing oligonucleotides oMW305, oMW306, oMW416, oMW417 and oMW421 using a previously described protocol (*Whitby and Dixon, 1998*). The junction was labelled with $^{32}$P at the 5′-terminus of one of its component oligonucleotides using polynucleotide kinase.

## In vitro cleavage assays

*S. pombe* Mus81-Eme1 and Mus81$^{DD}$-Eme1 were purified using an established protocol (*Gaskell et al., 2007*). Cleavage reactions, and their analysis by native (10% polyacrylamide) and denaturing (15% polyacrylamide) PAGE, have been described previously (*Gaskell et al., 2007*; *Osman et al., 2003*). The reaction buffer contained 25 mM Tris-HCl (pH 8.0), 1 mM dithiothreitol, 100 μg/ml bovine serum albumin, 6% glycerol and 10 mM MgCl$_2$. Reactions were started by the addition of protein and were incubated at 30°C for 30 min. Reactions were stopped by the addition of one-fifth volume of stop mix (2.5% SDS, 200 mM EDTA and 10 mg/ml proteinase K) and further incubation at 30°C for 15 min to deproteinize the mixture. Reaction products were processed and run on native and denaturing gels as described (*Gaskell et al., 2007*; *Osman et al., 2003*). When mapping cleavage sites on denaturing gels, reaction products were run alongside a Maxam-Gilbert G + A sequencing ladder of the relevant labeled oligonucleotide. A 1.5-base allowance was made to compensate for the nucleoside eliminated in the sequencing reaction. Gels were dried on 3 MM Whatman paper and analyzed with a Fuji FLA3000 PhosphorImager (Fujifilm Corp., Japan).

## Acknowledgements

We thank Joe Park for assistance with some of the recombination assays.

## Additional information

### Funding

| Funder | Grant reference number | Author |
| --- | --- | --- |
| Wellcome | 090767/Z/09/Z | Matthew C Whitby |
| EP Abraham Cephalosporin Trust Fund | | Matthew C Whitby |

The funders had no role in study design, data collection and interpretation, or the decision to submit the work for publication.

### Author contributions

CAM, MON, Formal analysis, Investigation, Methodology, Writing—review and editing; AF, INW, CB, Investigation, Writing—review and editing; FO, Formal analysis, Supervision, Investigation, Methodology, Writing—review and editing; MCW, Conceptualization, Formal analysis, Supervision, Funding acquisition, Methodology, Writing—original draft, Writing—review and editing

### Author ORCIDs

Matthew C Whitby, http://orcid.org/0000-0003-0951-3374

## Additional files

### Supplementary files

• Supplementary file 1. *Schizosaccharomyces pombe* strains.

• Supplementary file 2. Oligonucleotides.

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
