## [Decision Letter]

Thank you for submitting your article "Inter-fork Strand Annealing causes genomic deletions during the termination of DNA replication" for consideration by *eLife*. Your article has been reviewed by three peer reviewers, and the evaluation has been overseen by a Reviewing Editor and Kevin Struhl as the Senior Editor.

The reviewers have discussed the reviews with one another and the Reviewing Editor has drafted this decision to help you prepare a revised submission.

In this study, the authors investigate the mechanism of replication restart by homologous recombination in fission yeast. A directional replication fork barrier, RTS1, had been shown to stimulate recombination between repeat sequences located close to RTS1. Here, the authors investigated the effect of increasing the distance between repeats, either before or after RTS1 (RTS1 is inserted between direct repeats of ade6). Having the RFB in the "active" orientation stimulates inter-repeat recombination. Placing a spacer sequence between the repeats, especially past the RFB, increases recombination events, particularly those occurring by a mechanism that requires Rad52 but is independent of Rad51. Deletion formation was also suppressed in exo1 and mus81 mutants. The requirement for Mus81 supports the idea that deletions do not result from fork collapse and SSA. This spacer effect appears to depend on the probability that a converging fork will encounter the stalled fork. Genetic requirements suggest a model coined inter-fork strand annealing (IFSA) accounts for the spacer effect. One appealing aspect of the model is that it involves Rad52 annealing activity without suggesting that Rad52 promotes strand invasion.

It is becoming increasingly apparent that problems arising during DNA replication are potent drivers of genomic instability. Here the authors identify a mechanism for deletion formation at stalled forks that is distinct from SSA and relies on Rad52, Exo1 and Mus81. Given the current interest in how Rad52 and Mus81 facilitate replication fork restart in mammalian cells, the findings will be of interest to a wide audience.

Overall, the work represents a significant contribution and should be published in *eLife*. No additional experiments are required. However, the authors need to revise the manuscript to correct the following limitations of the manuscript:

1) The authors need to tone-down the discussion about IFSA being a "new mechanism": their statements to that effect are over-stated, and they disregard work on strand annealing during DNA replication in bacteria. The model is, in fact, a replication-slippage model that has been discussed for decades, with the added – important – feature that the target is in a separate replication fork. The authors need to credit prior precedents while acknowledging the novel aspects that implicate two replication forks.

2) Related to the model, as indicated, the single-strand derives from a resected regressed fork. However, an alternative, which they should acknowledge, is that the nascent 3' strand is simply unwound by helicases from the template.

3) The text from subsection “RF convergence is needed for DNA spacer-dependent deletions” is difficult to follow, and it needs to be rewritten to make their argument clearer.

4) It is difficult to guess whether IFSA might account for any genome instability under normal growth conditions – the authors need to address the frequency of physiological occurrence of IFSA. As far as the reviewers know, e.g., there are no orphan examples of natural or genotoxin-induced CNVs that are explained by IFSA. Hence, the authors need to discuss whether IFSA might explain a significant class of CNVs, natural or genotoxin-induced, or any other common genomic rearrangement.

---

## [Author Response]

*1) The authors need to tone-down the discussion about IFSA being a "new mechanism": their statements to that effect are over-stated, and they disregard work on strand annealing during DNA replication in bacteria. The model is, in fact, a replication-slippage model that has been discussed for decades, with the added – important – feature that the target is in a separate replication fork. The authors need to credit prior precedents while acknowledging the novel aspects that implicate two replication forks.*

We have re-written and expanded the Discussion to better acknowledge current understanding of the different mechanisms that can give rise to genomic deletions and other structural variants. In doing so, we have removed all mention of IFSA being a “new/novel mechanism”, and made reference to previous replication slippage models, as well as other replication-based mechanisms such as FoSTeS and MMBIR.

*2) Related to the model, as indicated, the single-strand derives from a resected regressed fork. However, an alternative, which they should acknowledge, is that the nascent 3' strand is simply unwound by helicases from the template.*

We have included two alternative models for IFSA in addition to the model presented in Figure 5 (Results, subsection “A new model: Inter-Fork Strand Annealing” and Figure 5—figure supplement 1). One model depicts the nascent 3' strand being unwound, and the other shows how annealing between lagging strand gaps could result in deletions.

*3) The text from subsection “RF convergence is needed for DNA spacer-dependent deletions” is difficult to follow, and it needs to be rewritten to make their argument clearer.*

We have completely re-written and extended this section to make the arguments about fork convergence clearer (Results, subsection “RF convergence is needed for SDDs”).

4) It is difficult to guess whether IFSA might account for any genome instability under normal growth conditions – the authors need to address the frequency of physiological occurrence of IFSA. As far as the reviewers know, e.g., there are no orphan examples of natural or genotoxin-induced CNVs that are explained by IFSA. Hence, the authors need to discuss whether IFSA might explain a significant class of CNVs, natural or genotoxin-induced, or any other common genomic rearrangement.

We have extended the Discussion to include more about the potential frequency and relevance of IFSA to the formation of chromosomal deletions and other structural variants in humans.